# Modeling human visual search: A combined Bayesian searcher and saliency map approach for eye movement guidance in natural scenes

**Melanie Sclar**[1*]
melaniesclar@gmail.com

**Gaston Bujia**[1 2*]
gbujia@dc.uba.ar

**Sebastian Vita**[1]
seba.vita@gmail.com

**Guillermo Solovey**[2]
gsolovey@gmail.com

**Juan E Kamienkowski**[1]
juank@dc.uba.ar

[1] Laboratorio de Inteligencia Artificial Aplicada, Instituto de Ciencias de la Computación, Universidad de Buenos Aires – CONICET, Argentina
[2] Instituto del Cálculo, Universidad de Buenos Aires – CONICET, Argentina

## Abstract

Finding objects is essential for almost any daily-life visual task. Saliency models have been useful to predict fixation locations in natural images, but they provide no information about the time-sequence of fixations. Nowadays, one of the biggest challenges in the field is to go beyond saliency maps to predict a sequence of fixations related to a visual task, such as searching for a given target. Bayesian observer models have been proposed for this task, as they represent visual search as an active sampling process. Nevertheless, they were mostly evaluated on artificial images, and how they adapt to natural images remains largely unexplored.

Here, we propose a unified Bayesian model for visual search guided by saliency maps as prior information. We validated our model with a visual search experiment in natural scenes recording eye movements. We show that, although state-of-the-art saliency maps are good to model bottom-up first impressions in a visual search task, their performance degrades to chance after the first fixations, when top-down task information is critical. Thus, we propose to use them as priors of Bayesian searchers. This approach leads to a behavior very similar to humans for the whole scanpath, both in the percentage of target found as a function of the fixation rank and the scanpath similarity, reproducing the entire sequence of eye movements.

## 1 Introduction

Visual search is a natural task that humans perform in everyday life, from looking for someone in a photograph to searching where you left your favorite mug in the kitchen. Finding our goal relies on our ability to gather visual information through a sequence of eye movements, performing a discrete sampling of the scene. The fixations of the gaze follow different strategies, trying to minimize the number of steps needed to find the target [1, 34, 38, 46]. This process is an example of active sensing, as humans perform decisions about future steps making inferences based on the information gathered so far to fulfill a task [45, 12]. Moreover, this decision-making behavior is goal dependent; for instance, whether it is task-driven or simple curiosity. Predicting eye movements necessary to meet a goal is a computationally-complex task: it must combine the bottom-up information capturing processes and the top-down integration of information and updating of expectations in each fixation.

---

[*]Indicates equal contribution. Correspondence should be addressed to M.S. or G.B.

A related task is the prediction of the most likely fixation positions in the scene to build a *saliency map*, identifying regions that draw our attention within an image [16]. Nevertheless, they produce accurate results only in the first few fixations of free exploration tasks [41] as they cannot make use of the sequential nature of the task nor combine information through the sequence of fixations. Thus, they may not be able to replicate these results when performing a complex task, like visual search. Recently, Zhang et al. [47] proposed an extension of those models to predict the sequence of fixations during visual search in natural images. The authors use a greedy algorithm, elaborating an attention map related to the search goal. Greedy algorithms implies using ad-hoc known behaviors of human visual search –like inhibition of return– that arise naturally with longer-sighted objective functions.

Nowadays, there is a growing interest in Bayesian models given their good results at modeling human behavior, and also for having straightforward interpretations related to human information processing [43]. For instance, different Bayesian Models were applied to decision making or perceptual tasks [29, 33, 36, 44, 42, 20, 39, 26]. For visual search tasks, Najemnik and Geisler [27] have proposed a very influential model called the *Ideal Bayesian Searcher (IBS)*. IBS decides the next eye movement based on its prior knowledge, a visibility map, and the current state of a posterior probability that is updated after every fixation. Recently, Hoppe and Rothkopf [14] proposed a visual search model that incorporates planning, looking more than one saccade ahead. As Najemnik and Geisler [27], their task uses artificial stimuli and is specifically designed for maximizing the difference between models, i.e finding the target in very few fixations on artificial stimuli. In order to extend these results to natural images, it is necessary to incorporate the information available in the scene.

Here, we show that an IBS model combined with state-of-the-art saliency maps as priors performs as well as humans in a visual search task on natural scenes images. Moreover, we incorporate a different update rule to the IBS model, based on a correlation for the template response. This modification could incorporate the effect of distractors. Previous work compares the general performance between humans and models (targets found and total number of fixations). Moving one step further, we quantitatively compare the scanpaths (ordered sequence of fixations) produced by the model with the ones recorded by human observers.

## 2    Visual search in natural indoor images: Human data

We set up a visual search experiment in which participants have to search for an object in a crowded indoor scene. The target initially appears in the center of the screen (subtending $144 \times 144$ pxs), which is replaced after 3 secs by a fixation dot in a pseudo-random position, at least 300 pxs away from the target position in the scene (Fig. S1). This was done to avoid starting the search close to the target. The initial position was the same for all participants in a given image. Moreover, the search image appears after the participant fixates the dot (Fig. S1). Thus, all observers initiate the search in the same place for a given image. The search image disappears when the participant fixates the target or after $N \in \{2, 4, 8, 12\}$ saccades. $N$ was randomized for each participant and image.

Indoor pictures with several objects, and no human figures or text, were selected from public databases (134 images in total). They were presented at a $1024 \times 768$ resolution (subtending $28.3 \times 28.8$ degrees of visual angle). For each image, the target was chosen by selecting a region of $72 \times 72$ pxs around it that was not repeated in the image –because we weren't evaluating the accuracy of memory retrieval–. See details on the paradigm and data acquisition and preprocessing in the Appendix.

Fifty-seven subjects participated in the visual search task (See Appendix). As expected, the proportion of targets found increases as a function of the saccades allowed (Fig. S2A), reaching a plateau from 8 to 12 saccades allowed and on (Fig. S2A and data from a preliminary experiment with up to 64 saccades not shown). Overall, eye movements recorded –saccade amplitude and direction, and fixation spatial distribution– behave as expected, also as function of fixation rank (Fig. S2B-D).

## 3    Searcher modeling: combining saliency maps with Bayesian searchers

### 3.1    Exploring saliency maps

Saliency maps are usually estimated in a scene-viewing task, where observers freely explore a set of images. As shown with flash-preview moving-window paradigms, even less than a few hundreds of milliseconds' glimpse of a scene can guide search, as long as sufficient time is subsequently available

to combine the prior knowledge with the current visual input [28, 41, 8]. Importantly, this is more relevant in the first saccades, and its predictive power decays with the search progresses [41].

With the goal of understanding which features guide the search, we choose and compare four different state-of-the-art saliency maps based on DNNs –DeepGaze2 (DG2) [23], MLNet [10], SAM-VGG and SAM-ResNet [11]– with a saliency model based on Gaussian filters, extracting purely low-level image information (ICF; Intensity Contrast Feature) [23]. We also include a baseline with just the center bias, modeled by a 2D Gaussian. As the control model, we built a human-based saliency map using the accumulated fixation position of all observers for a given image, smoothed with a Gaussian kernel of approximately one degree ($std = 25$ pxs). Given that observers were forced to begin each trial in the same position, we did not use the first fixations but the third, to capture the regions that attract human attention.

First, we evaluated how these models perform in predicting fixations along the search by themselves, considering each saliency map as a binary classifier on every pixel and using Receiver Operator Curves (ROC) and Area Under the Curve (AUC) to measure their performance (Fig. S3A). As expected, the human-based saliency map is superior to all other saliency maps, and the center bias map was clearly the worst (Fig. S3B). This is consistent with the idea that the first steps in visual search are mostly guided by image saliency. The rest of the models have similar performance on AUC, with DG2 performing slightly better than the others (Fig. S3B). Using different definitions of AUC [2, 32, 7, 24] showed the same trend (Table S2). If we consider all fixations the AUC is reduced for all models, including the human-based saliency map built on the third fixations (Fig. S3).

All models reached a maximum in AUC at the second fixation (Fig. S3C). Interestingly, the center bias begins at a similar level as the other but decays more rapidly, reaching 0.5 in the fourth fixation. Thus, other saliency maps must capture some other relevant visual information. Nevertheless, the AUC values from all saliency maps decay smoothly (Fig. S3C). This suggests that the gist the observers are able to collect in the first fixations is largely modified by the search. Top-down mechanisms must take control and play major roles in eye movement guidance as the number of fixations increase [15]. The DG2 model performed better over all fixation ranks (Fig. S3C).

## 3.2 Modifying the Ideal Bayesian Searcher (IBS) to handle natural scenes

Najemnik and Geisler [27]'s IBS computes the optimal next fixation location in each step. It considers each possible next fixation and picks the one that maximize the probability of correctly identifying the target's location after the fixation (See Appendix for details). IBS has only been tested in artificial images, where participants look for a small gabor patch among $1/f$ noise in one out of 25 locations. Our work is, to our knowledge, the first one to test this approach in natural scenes. Below, we discuss the modifications needed to apply to the IBS to model eye movements in natural images.

Firstly, the original IBS model had a uniform prior distribution. Since we are trying to model fixation locations in a natural scene, we introduced a saliency model as the prior. The $prior(i)$ will be the average of the saliency in the $i$-th grid cell. We restrict the possible fixation locations to be analyzed to the center points of a grid of $\delta \times \delta$ pixels, and collapsed consecutive fixations within a cell into one fixation to be fair with the model's behavior.

Secondly, the presence of the target in a certain position is not as straightforward as in artificial stimuli, where all the incorrect locations are equally dissimilar. In natural images there are often distractors, i.e. locations that are visually similar to the target, especially if seen with low visibility. Thus, we redefine the template response as $\tilde{W}_{ik(t)} \sim \mathcal{N}(\tilde{\mu}_{ik(t)}, \tilde{\sigma}^2_{ik(t)})$, with $\tilde{\mu}_{ik(t)}, \tilde{\sigma}_{ik(t)}$ as follows:

$$\tilde{\mu}_{ik(t)} = \mu_{ik(t)} \cdot \left(d'_{ik(t)} + \frac{1}{2}\right) + corr_i \cdot \left(\frac{3}{2} - d'_{ik(t)}\right) \in [-1, 1] \quad \text{and} \quad \tilde{\sigma}_{ik(t)} = \frac{1}{a \cdot d'_{ik(t)} + b} \quad (1)$$

$d'_{ik(t)}$ and $\tilde{W}_{ik(t)}$ are the visibility and template response at display location $i$ when the fixation is at display location $k(t)$, and $corr_i \in [-0.5, 0.5]$ is the cross-correlation of location $i$ and the target image, used as a measure of image similarity. Moreover, we modified $\sigma_{ik(t)}$ to keep the variance depending on the visibility, but we incorporate two parameters $a$ and $b$ that jointly modulate the inverse of the visibility and prevent $1/d'$ from diverging. Recently, Bradley et al. [3] simplified the task by fitting a visibility map built from first-principles with parameters estimated per participant. Here, we further simplified it by using a 2D Gaussian with the same parameters for every participant,

taken *a priori*, and therefore also avoiding a potential leak of information about the viewing patterns to the model.

We call this variation of the model *correlation-based IBS* (cIBS). The data, models, and code will be publicly available upon publication. To our knowledge, this is the first time that an implementation of Najemnik and Geisler [27] is publicly available, and it is largely optimized (See Appendix).

### 3.2.1 Evaluating searcher models on human data

We first evaluate the updating of probabilities and the decision rule of the next fixation position of the proposed cIBS model. We use IBS for comparison, in which the *template response* accounts only for the presence of the target, and not for the similarity of the given region with the target. Also, we implemented two other basic models: a *Greedy searcher* and a *Saliency-based searcher*. The *Greedy searcher* bases its decision on maximizing the probability of finding the target in the next fixation. It only considers the present posterior probabilities and the visibility map, and does not take into account how the probability map is going to be updated after that. The *Saliency-based searcher* simply goes through the most salient regions of the image, adding an inhibition-of-return effect to each visited region. In these models, we used DG2 as the prior, because it is the best performing saliency map of the previous section. We also evaluate the usage of different priors with cIBS, comparing with the center bias alone (a centered 2D Gaussian), a uniform distribution ("flat" prior), and a white noise distribution.

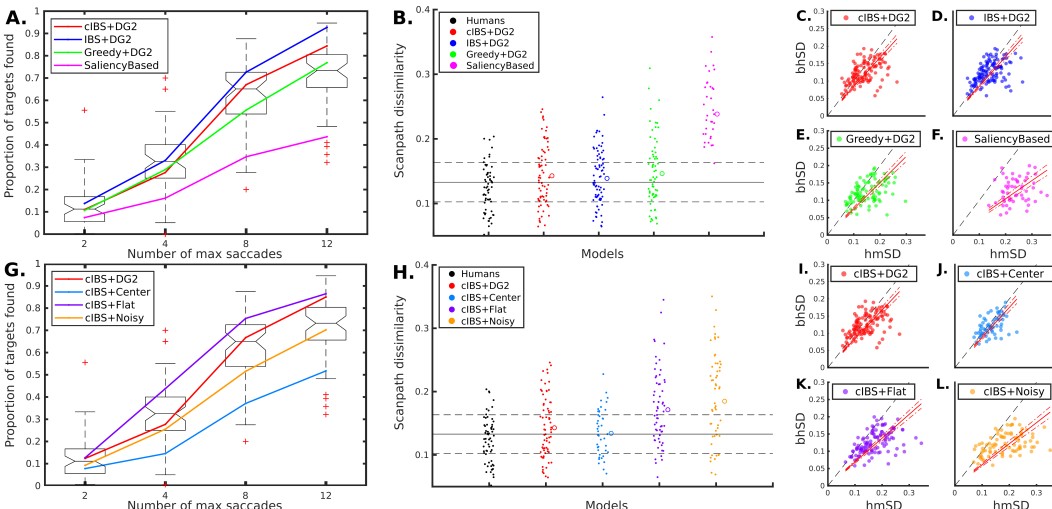

Figure 1: **A)** Proportion of target found by models (colored lines) and humans (boxes) according to the maximum number of saccades allowed. **B)** Distribution of scanpath similarity for each of the models and humans. **C-F)** Mean scanpath dissimilarity between humans (bhSD) as function of the mean scanpath dissimilarity between each human observer and a model (hmSD). Scanpath dissimilarity between different saliency priors with cIBS and humans. **G)** similar to A). **H)** similar to B). **I-L)** similar to (C-F). Only correct trials were considered in panels (b-f, h-l).

The proportion of targets found for any of the possible saccades allowed was used as a measure of overall performance (Fig. 1A, G). In Table 1 we summarize the metrics comparing humans and models: Weighted Distance measures the mean difference between curves (Fig. 1A, G), Jaccard index represents the proportion of targets found by the model to the total targets found by the subjects, and Mean Agreement measures the proportion of trials where subject and model had the same performance –both subject and model found (or not) the target– (See Appendix). When comparing different searchers with the same prior, cIBS has the best agreement with the humans' performance as a compromise of different metrics (Table 1). In particular, the performance of cIBS+DG2 is the closest to the human mean agreement ($0.64 \pm 0.086$). Nevertheless, the curves of the IBS and Greedy models were also very close to humans (Fig 1A) and each of them performed better in one metric (Table 1). It is important to note that Weighted Distance significantly improves when adding the distractor component (cIBS vs IBS). Only the basic Saliency-based searcher had poorer agreement with the human performance, showing that template matching weighted by visibility is a plausible

Table 1: Different measures on performance and scanpath similarity between humans and models. *Weighted distance* is the distance between humans and model performance weighted by humans dispersion. *Mean Agreement* and *Jaccard Index* measure coincidence of humans and model's correct target detections (See Appendix). $\rho$ quantifies the correlation between humans and model's scanpath similarities. *Linear Regression* corresponds to the slope in a simple $y \sim x$ model, between humans and models. Only correct trials were considered and averaged for each image ($N = 134$).

| Searcher: | cIBS | IBS | Greedy | Saliency | cIBS | cIBS | cIBS |
|---|---|---|---|---|---|---|---|
| Prior: | DG2 | DG2 | DG2 | DG2 | Center | Flat | Noisy |
| Weighted Distance | 0.31 | 0.78 | 0.13 | 2.14 | 1.38 | 0.72 | 0.15 |
| Mean agreement | 0.64 (0.084) | 0.64 (0.096) | 0.63 (0.082) | 0.55 (0.092) | 0.60 (0.086) | 0.60 (0.109) | 0.61 (0.082) |
| Jaccard Index | 0.51 (0.098) | 0.54 (0.105) | 0.47 (0.091) | 0.32 (0.073) | 0.38 (0.086) | 0.51 (0.109) | 0.46 (0.089) |
| Linear regression (slope) | 0.84 | 0.85 | 0.76 | 0.52 | 0.88 | 0.69 | 0.61 |
| Spearman Correlation ($\rho$) | 0.50 | 0.48 | 0.44 | 0.22 | 0.50 | 0.54 | 0.43 |

mechanism for searching potential targets. Then, we explored the importance of the prior, comparing the best searcher with the chosen prior against the different basic priors. Again, cIBS+DG2 had better overall agreement with humans' behavior (Fig. 1G and Table 1) and, interestingly, is the only model that presented a step-like function characteristic from humans (Fig. 1G).

Going one step further, we compare the scanpaths using the *scanpath dissimilarity* measure proposed by Jarodzka et al. [18]. Briefly, the scanpath is defined as a sequence of fixations $u_i$ (*(x,y)-coordinates*) where the $i$-th saccade is the shortest path (vector) going from $u_i$ to $u_{i+1}$. Each $u_i$ may not be exactly a fixation, but the center of a cluster of several fixations, making this measure more robust. Depending on the objective of the comparison, it could be used with different summary measures. As we aim to compare the sequence of explored locations between humans and our model, we use the shape of the scanpath. It is calculated as the $[0, 1]$ normalized difference between the saccade vectors (lower is better).

Comparing between searchers, we observed that both cIBS and IBS were almost indistinguishable from humans, but the Greedy searcher was still very close. Only the Saliency-based searcher resulted in a significantly different behavior (Fig. 1B-F). We quantified this relation by measuring the correlation and the slope of a linear regression (with null intercept) of the dissimilarity between humans (bhSD) and between humans and the model (hmSD) (Table 1). The rationale of these measures is that the ground truth of each image (the human scanpath) has different variability across images, and we cannot expect that in the case of very diverse scanpaths (higher dissimilarity) among humans, the model was close to all of them. All correlations and linear models presented a small p-value ($p < 10^{-4}$ for correlations and $p < 10^{-10}$ for linear models), except for control models. A close look at the correlation between the dissimilarity measure in humans and models evidenced that there were only a small fraction of images that departed from the human's scanpaths (Fig. 1C-F, see also Fig. S5).

When comparing between priors, we observed that both the models with DG2 and Center priors were closer to humans' values, and the flat and noisy priors had larger dissimilarities (Fig 1H). The model with a flat prior had a slightly better correlation, but both the models with DG2 and Center priors had good correlation and slopes closer to 1 (Table 1). This suggests that the initial center bias is a fair approximation of human priors. Nonetheless, although both scanpaths were indistinguishable from humans, saliency adds information that makes the model with DG2 quite better in finding the target.

## 4   Conclusions

Previous efforts on including contextual information aimed mainly to predict image regions likely to be fixated. For instance, they statically combined a spatial filter-based saliency map with previous knowledge of target object positions on the scene [41]. Some other works aimed to predict the sequence of fixations, but efforts on non-Bayesian modeling mainly used greedy algorithms [30, 47]. In contrast, as we and others have shown, a more long-sighted objective function has the additional benefit of having some known behaviors of human visual search arise naturally [27, 14]. For example, Zhang et al. [47] forced inhibition of return, while in our model has it implicitly incorporated. Crucially, Bayesian frameworks are also highly interpretable and connect our work to other efforts in modeling top-down influences in perception and decision-making from first principles. For instance,

Bruce and Tsotsos [5, 6] proposed a saliency model based on information maximization principle, which demonstrated great efficacy in predicting fixation patterns across both pictures and movies. Also, search models had been implemented for a fixed-gaze task, for instance, to deal with the reliability of visual information across items and displays [25].

Although Najemnik and Geisler [27]'s model was a very insightful and influential proposal, our work is, to our knowledge, the first one that uses a Bayesian framework to predict eye movements during visual search in natural images. It is a leap in terms of applications since prior work on Bayesian searchers was done in very constrained artificial environments [27]. We addressed possible modifications when considering the complexities of natural images. Specifically, the addition of a saliency map as prior, the modification of the template response's mean, and shift in visibility. Finally, we share our optimized code to replicate both Najemnik and Geisler [27] and our results.

Finally, the present work expands the growing notion of the brain as an organ capable of interacting with novel, noisy and cluttered scenarios through Bayesian inference, building complete and abstract models of its environment. Nowadays, those models cover a broad spectrum of cognitive functions, such as decision making and confidence, learning, perception, and others [20, 39, 26].

## Acknowledgments and Disclosure of Funding

### Author contributions

M.S. prepared the dataset and coded the first implementation of the presented models, including numerical optimizations and speed-ups. M.S., G.S. and J.K. designed the task, collected the human data and defined the model idea. G.B. and S.V. pruned the model's code, extended it and explored their parameters. M.S., G.B., S.V. and J.K. performed the analysis. The manuscript was written by G.B., M.S. and J.K.

### Acknowledgments

We thank P Lagomarsino and J Laurino for their collaboration with the data acquisition, C Diuk (Facebook), A Salles (OpenZeppelin) and Matias J Ison (Univ. of Nottingham, UK) for their feedback and insight on the work, and K Ball (UT Austin) for English editing of this article. The authors were supported by the CONICET and the University of Buenos Aires (UBA). The research was supported by the UBA (20020170200259BA), the ARL (Award W911NF1920240) and the National Agency of Promotion of Science and Technology (PICT 2016-1256).

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

# A Visual search in natural indoor images: Paradigm and human data acquisition and preprocessing

## A.1 Participants

Fifty-seven subjects (34 male; age $25.1 \pm 5.9$ years old) participated in the visual search task. All subjects were naïve to the objectives of the experiment, and had normal or corrected-to-normal vision and provided written informed consent according to the recommendations of the declaration of Helsinki to participate in the study.

## A.2 Paradigm and procedure

We set up a visual search experiment in which participants have to search for an object in a crowded indoor scene. First, the target is presented in the center of the screen, subtending $144 \times 144$ pixels of visual angle (Fig. S1). After 3 seconds, the target is replaced by a fixation dot at a pseudo-random position (Fig. S1) the search image appears after the participant fixates the dot, and it is presented at a $768 \times 1024$ resolution (subtending $28.3 \times 28.8$ degrees of visual angle) (Fig. S1).

The program automatically detects the end of each saccade during the target search. This period finishes when the participant fixates the target or after $N$ saccades, allowing an extra 200ms for the participant to be able to process the information in that last fixation [21]. The maximum number saccades allowed ($N$) varied between 2 (13.4% of the trials), 4 (14.9%), 8 (29.9%) or 12 (41.8%) for most of the participants. These values were randomized for each participant, independently of the image.

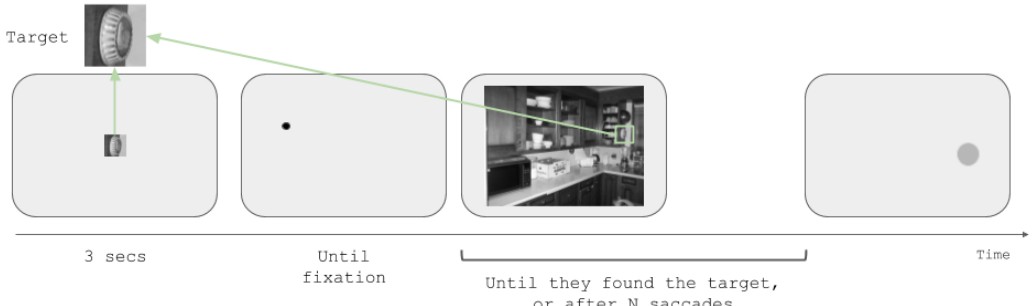

Figure S1: Paradigm schema

After each trial, the participants are forced to guess the position of the target, even if they had already found it. They are instructed to cover the target position with a Gaussian blur, first by clicking on the center and then by choosing its radius. This is done by showing a screen with only the frame of the image and a mouse pointer –a small black dot– to select the desired center of the blur (Fig. S1). When choosing a position with the mouse, a Gaussian blur centered at that position is shown, and the participants are required to indicate the uncertainty of their decision by increasing or decreasing the size of the blur using the keyboard. Position and uncertainty reports were not analyzed in the present study.

A training block of 5 trials was performed at the beginning of each session with the experimenter present in the room. After the training block, the experiment started and the experimenter moved to a contiguous room. The images were shown in random order. Each participant observed the 134 images in three blocks. Before each block, a 9-point calibration was performed and the participants were encouraged to get a small break to allow them to rest between blocks. Moreover, each trial starts with the built-in drift correction procedure from the EyeLink Toolbox, in which the participant has to fixate in a central dot and hit the spacebar to continue. If the gaze is detected far from the dot, a beep signaled the necessity of a re-calibration. The experiment was programmed using PsychToolbox and EyeLink libraries in MATLAB [4, 19].

### A.3 Stimuli

We collected 134 indoor pictures from Wikimedia commons, indoor design blogs, and LabelMe database [35]. The selection criterion was that scenes should have several objects and no human figures or text should be present. Moreover, the images are in black and white to make the task take more saccades, since color is a very strong bottom-up cue. Also, a pilot experiment with 5 participants was performed to select images that usually require several fixations to find the target. The original images were all larger or equal than $1024 \times 768$ pixels, and all were cropped and/or scaled to $1024 \times 768$ pixels. For each image, a single target was manually selected among the objects of $72 \times 72$ pixels or less that were not repeated in the image –because we weren't evaluating the accuracy of memory retrieval–. For all targets, we considered a surrounding region of $72 \times 72$ pixels.

### A.4 Data acquisition

Participants were seated in a dark room, 55 cm away from a 19-inch Samsung SyncMaster 997MB monitor (refresh rate = 60Hz), with a resolution of $1280 \times 960$. A chin and forehead rest was used to stabilize the head. Eye movements were acquired with an Eye Link 1000 (SR Research, Ontario, Canada) monocular at 1000 Hz.

### A.5 Data preprocessing

The saccade detection was performed online with the native EyeLink algorithm with the default parameters for cognitive tasks. Fixations were collapsed into a grid with cells of $32 \times 32$ pixels, resulting in a grid size of $32 \times 24$ cells. We explored the size of the grid in terms of model performance. Consecutive fixations within a cell were collapsed into one fixation to be fair with the model behavior. Also, fixations outside the image region were displaced to the closest cell. As we considered fixations, blinks periods were excluded.

The trial was considered correct (target found) if the participant fixated into the target region ($72 \times 72$ pixels). Only correct trials were analyzed in terms of eye movements.

## B  Human behavior results

During the experiment, observers have to search for a given target object within natural indoor scenes. The trial stops when the observers find the target or after $N$ saccades ($N = 2, 4, 8, 12$). As expected, the proportion of targets found increases as a function of the saccades allowed (Fig. S2A), reaching a plateau from 8 to 12 saccades allowed and on (Fig. S2A and data from a preliminary experiment with up to 64 saccades not shown).

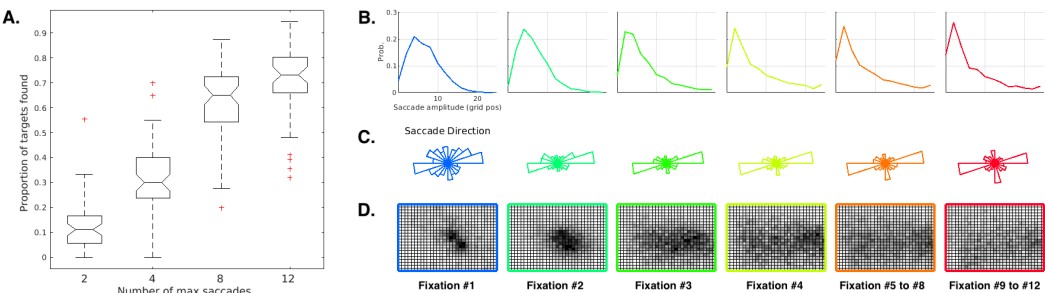

Figure S2: Experimental design and general behavior. **A)** Proportion of targets found as a function of the number of saccades allowed. Distributions of **B)** saccade length, **C)** saccade direction (measured in degrees from the positive horizontal axis), and **D)** fixations' position for different fixation ranks (1,2,3,4,5-8,9-12).

Overall, eye movements recorded behave as expected. First, the amplitude decreases with the fixation rank, presenting the so-called coarse-to-fine effect (Fig. S2B), and the saccades tended to be horizontal more than vertical (Fig. S2C). Finally, the initial spatial distribution of fixations had a central bias, and then extended first over the horizon until it covered the whole image, as the

targets are uniformly distributed along the scene (Fig. S2D). This effect could be partially due to the organization of the task (the central drift correction and presentation of the target), the setup (the central position of the monitor with respect of the eyes/head), and the images (the photographer typically centers the image); it also could be due to processing benefits, as it is the optimal position to acquire low-level information of the whole scene or to start the exploration.

## C   Exploring Saliency Maps

The first saliency models were built based on computer vision strategies, combining different filters over the image [16]. Some of these filters could be very general, such as a low-pass filter that gives the idea of the horizon [17, 40], or more specific, such as detecting high-level features like faces [9]. In recent years, deep neural networks (DNNs) have advanced the development of saliency maps. Many saliency models have successfully incorporated pre-trained convolutional DNNs in order to extract low and high-level features of the images [23, 10, 11]. These novel approaches were summarized in MIT/Tuebingen's collaboration website [22].

With the purpose of understanding which features guide the search in this section, we choose and compare five different state-of-the-art saliency maps for our task: DeepGaze 2 [23], MLNet [10], SAM-VGG and SAM-ResNet [11], and ICF (Intensity Contrast Feature) [23]. All the saliency models considered (except for ICF) are based on neural network architectures, using different convolutional networks (CNN) pretrained on object recognition tasks. These CNNs played the role of calculating a fixed feature space representation (feature extractor) for the image which then will be fed to a predictor function (in the models we consider, also a neural network). DeepGaze uses a VGG-19 [37] as feature extractor, and the predictor is a simpler four layers' CNN [23]. The MLNet model uses a modified VGG-16 [37] that returns several feature maps, and a simpler CNN is used as a predictor that incorporates a learnable center prior [10]. Finally, SAM could use both VGG-16 and ResNet50 [13] as two different feature extractors, and the predictor is a neural network with attentive and convolutional mechanisms [11]. ICF has a similar architecture to DeepGaze, but it uses of Gaussian filters instead of a neural network. This way, ICF extracts purely low-level image information (intensity and intensity contrast).

As the control model, we built a human-based saliency map using the accumulated fixation position of all observers for a given image, smoothed with a Gaussian kernel of approximately 1 degree (st. dev. = 25 pxs). Given that observers were forced to begin each trial in the same position, we did not use the first fixations but the third. This way we capture the regions that attract human attention.

### C.1   Prediction of observers' fixation positions

We evaluated how just the saliency models perform in predicting fixations along the search by themselves. Thus, we considered each saliency map **S** as a binary classifier on every pixel and used Receiver Operator Curves (ROC) and Area Under the Curve (AUC) to measure their performance. This comes with the difficulty that there is not a unique way of defining the false positive rate (**fpr**). In dealing with this problem, previous work on this task has used many different definitions of (ROC and its corresponding) AUC [2, 32, 7, 24]. Briefly, to build our ROC we considered the true positive rate (**tpr**) as the proportion of saliency map values above each threshold at fixation locations and the **fpr** as the proportion of saliency map values above threshold at non-fixated pixels (Fig. S3A).

As expected, the saliency map built from the distribution of third fixations performed by humans (human-based saliency map) is superior to all other saliency maps, and the center bias map was clearly worse than the rest of them (Fig. S3B). This is consistent with the idea that the first steps in visual search are mostly guided by image saliency. The rest of the models have similar performance on AUC, with DeepGaze2 performing slightly better than the others (Fig. S3B).

All models reached a maximum in AUC values at the second fixation except the human-based model that peaked at the third fixation as expected (Fig. S3C). Interestingly, the center bias begins at a similar level as the other saliency maps but decays more rapidly, reaching $0.5$ in the fourth fixation. Thus, other saliency maps must capture some other relevant visual information. Nevertheless, the AUC values from all saliency maps decay smoothly (Fig. S3C). This suggests that the gist the observers are able to collect in the first fixations is largely modified by the search. Top-down mechanisms must take control and play major roles in eye movement guidance as the number of fixations increase [15].

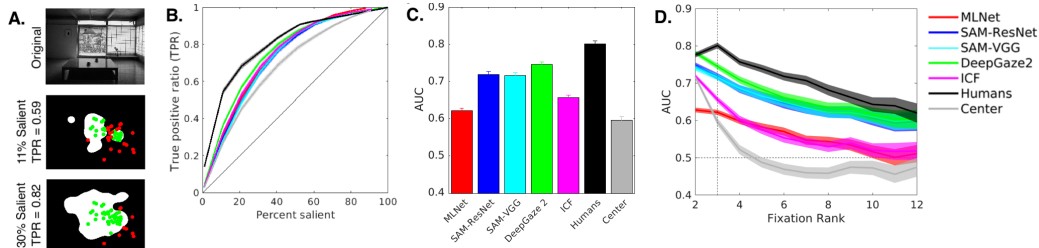

Figure S3: Saliency maps. **A)** Example on how to estimate the TPR for the ROC curve, **B)** ROC curves and **C)** AUC values for the third fixation and **D)** AUC for each Model as a function of the current Fixation Rank. Color mapping for models is consistent over B, C, and D.

The DeepGaze 2 model performed better over all fixation ranks, becoming indistinguishable from human performance in the second fixation (Fig. S3C).

## C.2   Comparing with other definitions of AUC

Different definitions of ROC-AUC found in `https://saliency.tuebingen.ai/` showed the same trend [2, 32, 7, 24]. All those ROC curves are built base on the idea of considering the saliency map as a binary classifier by applying a threshold. Here, we report three of them: AUC-Judd, AUC-Borji, and shuffled-AUC (or sAUC). These metrics differ mainly on the definition of the true positive rate and the false positive rate for the corresponding ROC curves. AUC-Judd considers human fixations as ground truth and all non-fixated pixels as negative cases. This way, the true positive rate is the proportion of pixels with saliency values above a certain threshold that were fixated. The false positive (fp) rate is the proportion of pixels with saliency values above a certain threshold that were not fixated. AUC-Borji keeps the same definition of the true positive rate, but uses a uniform random sample of image pixels as negatives and defines the saliency map values above a certain threshold at these pixels as false positives. Thus, the false positive rate is the proportion of those cases that were not fixated. Finally, sAUC is similar to AUC-Borji, instead of sampling pixels from the same image to define the fpr, it samples over fixation's locations on other images.

Table S2: Saliency maps: Different AUC metrics estimated for the saliency maps on the third fixation (Fig. S3)

| Saliency Maps | AUC-Judd | AUC-Borji | sAUC |
|---|---|---|---|
| MLNet | 0,7464 | 0,6797 | 0,6008 |
| SAM-VGG | 0.7321 | 0,6305 | 0,5666 |
| SAM-ResNet | 0,7339 | 0,6501 | 0,5820 |
| DeepGaze 2 | 0,7637 | 0,6537 | 0,5883 |
| ICF | 0,7509 | 0,7078 | 0,5808 |
| Humans | 0,8076 | 0,7792 | 0,7727 |
| Center | 0,6866 | 0,6739 | 0,5208 |

## C.3   Prediction of all fixation locations

If we consider all fixations the AUC is reduced for all models, including the human-based saliency map built on the third fixations (Fig. S4).

# D   Exploring Searcher models

## D.1   Description of the Ideal Bayesian searcher (IBS)

Najemnik and Geisler [27]'s IBS computes the optimal next fixation location in each step. It considers each possible next fixation and picks the one that will maximize the probability of correctly identifying the location of the target after the fixation. The decision of the optimal fixation location at step $T + 1$,

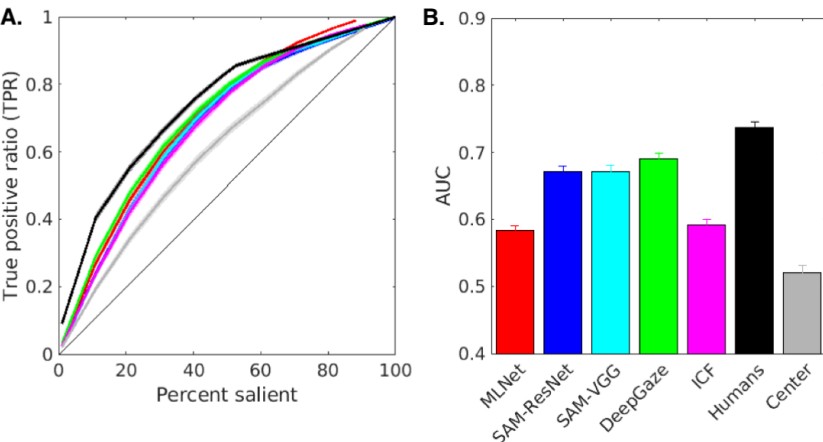

Figure S4: Saliency maps. **A)** ROC curves and **B)** AUC values for all fixations. Color mapping for models is consistent with Fig. S3.

$k_{opt}(T + 1)$, is computed as (eq. 2):

$$k_{opt}(T + 1) = argmax_{k(T+1)} \sum_{i=1}^{n} p_i(T)p(C|i, k(T + 1)) \tag{2}$$

where $p_i(T)$ is the posterior probability that the target is at the $i$-th location within the grid after $T$ fixations and $p(C|i, k(T + 1))$ is the probability of being correct given that the true target location is $i$, and the location of the next fixation is $k(T + 1)$. $p_i(T)$ involves the prior, the visibility map ($d'_{ik(T)}$) and a notion of the target location ($W_{ik(t)}$):

$$p_i(T) = \frac{prior(i) \cdot \prod_{t=1}^{T} exp\left(d'^2_{ik(t)} W_{ik(t)}\right)}{\sum_{j=1}^{n} prior(i) \cdot \prod_{t=1}^{T} exp\left(d'^2_{jk(t)} W_{jk(t)}\right)} \tag{3}$$

The template response, $W_{ik(t)}$, quantifies the similarity between a given position $i$ and the target image from the fixated position $k(t)$ ($t$ is any previous fixation). It is defined as $W_{ik(t)} \sim \mathcal{N}(\mu_{ik(t)}, \sigma^2_{ik(t)})$ where $\mu_{ik(t)} = \mathbb{1}_{(i = target\ location)} - 0.5$ and $\sigma_{ik(t)} = \frac{1}{d'_{ik(t)}}$. Abusing notation, in eq. 3, $W_{ik(t)}$ refers to a value drawn from this distribution.

### D.1.1  Why does IBS implicitly achieves inhibition of return?

Inhibition of return on the location $i$ is achieved because, assuming that the display location $i$ was already observed and the target was not found there, we can deduce that $\mu_{ik(t)} = -0.5$, $k(t) = i$ for some $t \leq T$, and therefore $W_{ik(t)} = W_{ii} \in \mathcal{N}\left(-0.5, \frac{1}{d'^2_{ii}}\right)$. As the visibility is maximum when the display location is the same as the one currently being observed, it follows that $W_{ii}$ will have little variance around its expected value of $-0.5$. This will imply that $exp\left(d'^2_{ii} W_{ii}\right) > 0$ is a small quantity, resulting in $p_i(T)$ being negligible. A similar intuition can be applied to display locations close to $i$, since they will still have a high degree of visibility.

### D.2  Description of the correlation-based Ideal Bayesian searcher (cIBS)

We restrict the possible fixation locations to be analyzed to the center points of a grid of $\delta \times \delta$ pixels, since it would be computationally intractable, and ineffective, to compute the probability of fixating in every pixel of a $1024 \times 768$ image.

The parameters of the 2D Gaussian model of the visibility map were chosen *a priori* estimated from values reported in Najemnik and Geisler [27], Bradley et al. [3], and were the same for for every participant. The bivariate Gaussian $\mathcal{N}(\mu, \Sigma)$ is centered on each fixation point ($\mu$ is the 2D-coordinate in pixels), and its covariance is $\Sigma = \left(\begin{smallmatrix} 2600 & 0 \\ 0 & 4000 \end{smallmatrix}\right) pxs^2$.

We define the template response as shown in eq. 1, where the two parameters $a$ and $b$ jointly modulate the inverse of the visibility and prevent $1/d'$ from diverging. These parameters were not included in the original model probably because $d'$ was estimated empirically (from thousands of trials and independently for each subject) and the $d'$ was never exactly equal to zero. We chose the parameters of the model using a classical grid search procedure in a previous experiment with a smaller dataset and the same best parameters ($\delta = 32$, $a = 3$ and $b = 4$) are used for all the models.

The intuition behind the implicit modeling of inhibition of return still holds with cIBS' new template response definition shown in eq. 1, but with the added disturbance of distractors. Now, if a display location is fairly visible, but the location is visually similar to the target location, the expected value of the template response will be higher than before: previously, it would have been $-0.5$ regardless of visual similarity.

### D.3 Metrics on the Comparison of Performances between Humans and Models

We used three measures to compare the performance (i.e. the probability of detecting a target) of each model with the human participants. Each of them focused on a slightly different aspect.

#### D.3.1 Distance weighted by the number of saccades allowed

In order to directly compare the performance curve of each model with human participants, for each possible number of saccades allowed $N \in \{2, 4, 8, 12\}$, we calculate the difference between the mean proportion of targets found by participants and by the model $m$ ($P_{subj}(N)$ and $P_m(N)$ respectively). For each number of saccades allowed, the difference is weighted by the standard deviation across participants $\sigma$. Then, the weighted distance $WD(m)$ is the mean value each of those values:

$$WD(m) = \sum_{N \in \{2,4,8,12\}} \frac{|P_{subj}(N) - P_m(N)|}{4\sigma^2} \tag{4}$$

#### D.3.2 Jaccard Index

Jaccard Index is a metric that allows us to measure the proportion of targets found by humans that are explained by the models. We represent each participant and each model as a boolean $S$-dimensional vector with a one in the $i$-th position if they found the target in the $i$-th image, and zero otherwise. $S$ is the number of images: in our experiment, $S = 134$.

Every participant has the same proportion of images with each maximum saccade possible $N$ (13.4% of the trials with $N = 2$, 14.9% with $N = 4$, 29.9% with $N = 8$ or 41.8% with 12). Nonetheless, the subset that gets each $N$ is chosen uniformly at random for each subject. This way, for each image we have subjects that were interrupted after 2, 4, 8, and 12 saccades. As each participant has a different sample of maximum saccades allowed across images, we apply these same constraints to the model. That is, when we want to compare with participant $p$, we apply their constraints to our model $m$. Then, we decide for each image if it can find the target in fewer saccades than the allowed. Given a model $m$, we define $sacc(m, i)$ as the number of saccades that the model needs to find the target in the $i$-th image. Then, for each participant we have $max\_sacc(i, p)$ as the number of saccades allowed for each image $i$ and participant $p$, with $p = 1 \ldots 57$ in our data. From these values, we construct the vector of targets found by the model using the saccade threshold distribution for each participant $p$, called $TFM_p^{(m)}$ (Targets Found by Model). $TFM_p^{(m)} \in \{0, 1\}^S$ is defined as:

$$TFM_p^{(m)}(i) = \begin{cases} 1 \text{ if } sacc(m, i) \leq max\_sacc(i, p) \\ 0 \text{ otherwise} \end{cases} \tag{5}$$

Each participant $p$ is also represented as a $S$-dimensional vector $TFP_p \in \{0, 1\}^S$ (Targets Found by Participant):

$$TFP_p(i) = \begin{cases} 1 & \text{if participant } p \text{ found the target in the } i\text{-th image} \\ 0 & \text{otherwise} \end{cases} \tag{6}$$

Then, we compute the Jaccard Index [31] between those vectors (7).

$$jaccard(p, m) = \frac{TFP_p \cap TFM_p^{(m)}}{TFP_p \cup TFM_p^{(m)}} \qquad (7)$$

### D.3.3  Mean Agreement

Another measure we considered to compare a model performance against humans was what we called the Mean Agreement Score (MAS), inspired by Mean Absolute Error. This measure calculates the mean proportion of trials where both the participant and the model had the same performance. This metric has the purpose of measuring the compromise between our model and the participants in their performance. We compute the difference between the boolean vectors like in Jaccard Index (6) and calculate the mean. Finally the Mean Agreement Score between model $m$ and participant $p$ is 1 minus that value:

$$MAS(x, y) = 1 - mean(|x - y|) \qquad (8)$$

### D.3.4  Scanpaths Examples

The small fraction of images that departed from the human's scanpaths (Fig. 1C-F; only $1\%$ using $\mu + 3\sigma$ for cIBS+DG2) correspond to cases where few people found the target or, interestingly, where there are different possible scanpath behaviors (i.e. some people start looking for a cup on the cupboard and others on the table) (Fig. S5). Further studies should be performed to explore individual differences between observers.

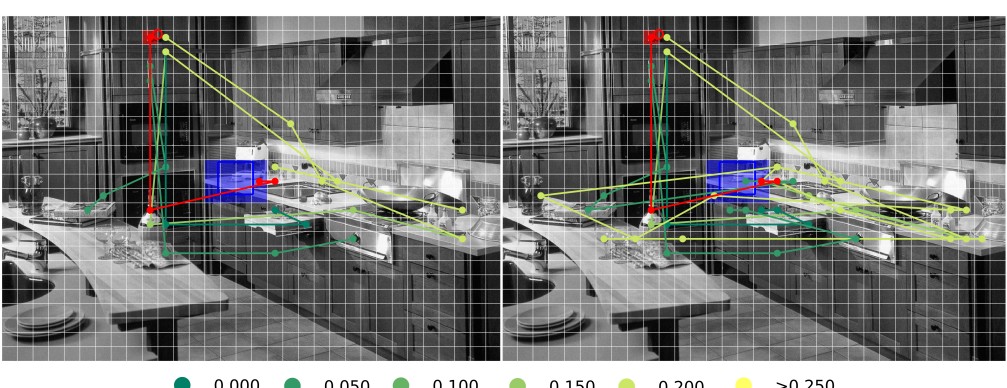

Figure S5: Scanpath prediction comparison. The figure is meshed by a fixed grid of $\delta = 32$px. Each curve represents a scanpath, in red the cIBS+DG2 model's scanpath and six scanpaths of participants colored according its own dissimilarity to the model scanpath. Left panel shows the first four fixations of each scanpath, and the right panel shows the whole scanpaths. The search target is represented with the blue square and the approximated first fixation with the red square. Above the image the hmSD and bhSD for this trial are reported. Image taken from *Wikipedia Commons.*

Some images showed an overall disagreement but, looking a little bit deeper, we can see that participants performed two different but consistent patterns. As the present implementation of the model is deterministic, it chose only one of those patterns (Fig. S5). In Figure S5, we show some of the human scanpaths and the model (cIBS+DG2) scanpath for one image to illustrate that specific behavior. In this case, the cup is the search target and there are two surfaces where, a priori, is equally likely to find it. We selected six human scanpaths with different hmSD values to show how the initial decision determines the behavior and the overall hmSD for that scanpath (Fig. S5, left panel), but almost all participants end up exploring both regions (Fig. S5, right panel). Note that dark green traces are scanpaths very similar to the model, while yellow traces are scanpaths that differ from the model. Further development of the should focus on mimicking these individual differences in visual search.

# E   Data and Code Availability

The model was fully developed in MATLAB. Saliency map calculations were performed using the public code provided by its respective authors [10, 11, 23].   All code, data, and parameters needed to reproduce this paper's results and visualizations will be available at `https://github.com/gastonbujia/VisualSearch`.

