# OpenReview forum: "Modeling human visual search: A combined Bayesian searcher and saliency map approach for eye movement guidance in natural scenes"
_NeurIPS.cc/2020/Workshop/SVRHM — SVRHM@NeurIPS Oral_

### Official Review · AnonReviewer3 · 2020-10-22
**Overall it is a novel bayesian searcher model that incorporates saliency as prior and accounts complexity of natural scenes**

**Rating:** 9
**Confidence:** 4

**Review:**

Intro:
Pros: good intro with relevant literature.
Cons: could have included more background in why human make eye movements (foveation)

Human Data:
Pros:
1. good control of search procedure starting and limitation of the number of saccades
2. comprehensive explanation of different evaluation methods
Cons:
1. not sure how they chose the search starting for each image. Is the distance between the starting point to the target consistent across different images?
2. all results missing stats report (p values).

Searcher Model:
Pros:
1. good comparisons and clear explanations for four different searcher models
2. using saliency map as prior for the natural images gives good intuition
3. It is a good point to introduce similarity measurement to account for distractors in the natural images
4. adding valuable contribution to open source code
5. clear explanation of the model equations
6. might be even more interesting to incorporate foveation to the model while making the fixation

Cons:
1. Figure 1G in 3.2.1. the notations of prior distributions are not consistent with the names refered in the text. It would be better to keep them consistent.
2. FIgure 1G in 3.2.1. Is it that cIBS+Flat refers to cIBS+uniform distribution is actually better than using DG2 as prior? A bit more elaborations on that would be helpful.

---

### Official Review · AnonReviewer2 · 2020-10-28
**Review for Modeling human visual search: A combined Bayesian searcher and saliency map approach for eye movement guidance in natural scenes**

**Rating:** 7
**Confidence:** 3

**Review:**

This aim of this work is to predict sequences of fixations in natural images during visual search using a Bayesian model. Compared to previous attempts, the novelty here saliency maps produced by from other models such as DeepGaze2 are sued as priors for the Bayesian model. This is also the first application of Ideal Bayesian Searchers to natural images. The authors validate this model by comparing with human eye movements during a visual search task. Interestingly, the saliency maps by themselves are sufficient to predict the first few saccades, but the Bayesian model is better for predicting later saccades. This suggests that the first few saccades in visual search are well modelled by bottom-up saliency maps, but a top-down component, modelled here in the Bayesian framework, is needed for later saccades, which depend on the task at hand.

I find this work interesting, and it fits the SVRHM workshop well. The experiments are well done and, as far as I can see, all the relevant controls have been performed. Therefore, I suggest accepting it. I have one major comment and a few minor suggestions to potentially improve the paper.

Major comment:
The conclusion that the first few saccades in visual search are well modelled by bottom-up saliency maps, but a top-down component is needed to model later saccades is well supported by the results. However, the current analysis is not able to really show differences between different versions of the Bayesian model. All the models look quite similar and I wonder if a different measure or a more in depth statistical analysis could help here. For example, the authors claim that “[cIBS+DG2] is the only model that presented a step-like function characteristic from humans”. I don’t see this in the graph. Without better statistical/experimental support, it would seem that cIBS+DG2, IBS+DG2 and cIBS+center all show this step-like function.

Minor suggestions:
-	The acronym “IBS” is used on line 49, but is only explained on line 105. I would suggest explaining it up front on line 49.
-	For future work, it may be interesting to use a “perceptual loss”, as used in GANs for example, instead of correlation for the template response. Essentially, this means comparing the internal representation of a pre-trained DNN layer to measure how similar a patch is to the target. Since correlation is *very* low-level, this may yield more human-like models.
-	Line 192: why does this model “implicitly incorporate inhibition of return”?
-	Typo in line 190: have shown.

---

### Official Review · AnonReviewer1 · 2020-10-31
**good introduction and analysis - needs more intuitive explanation in certain sections**

**Rating:** 8
**Confidence:** 4

**Review:**

Evaluation of quality:
-  good quantitative analysis and good details in the supplementary material.
- it seems that the sample size
- some improvement needed in writing (See below)
- dataset size: please add some comments on noise levels / quality of data in the main paper

Evaluation of clarity:
-  my main comment for improving clarity is that: the notations are sometimes referenced without introducing / providing an intuition. E.g., line 65: n=134.  Eq. 1 seems the key to understanding the approach but it is not clear what each of the entities mentioned in this equation stand for. This prevents the reader from getting  a clear assessment of the approach.
- typically in saliency prediction, the blur kernel size for fixation maps is chosen based on the rule that standard deviation of the Gaussian blur kernel is approximately equal to 1 degree viewing angle. What is the justification for std= 25pxs (line 85)?
- in the tables, please provide indication of best-performing method for ease of interpretation.

Evaluation of originality:
- the contribution of the paper is in providing two improvements over Najemnik and Geisler's (NG's) IBS to handle natural scenes (1: different prior, 2: different template response). To help understand the importance of both improvements, it would be good to show performance with just one improvement applied at a time. For example, what is the performance with the original prior and the novel template response? and with new prior but original template response? This test would help clearly see the contribution of the two improvements.

---

### Public Comment · ~Melanie_Sclar1 · 2020-12-08
**Response to reviewers**

We thank the reviewers for their positive comments and insights. We clarified the issues raised by the reviewers in the camera-ready version. Moreover, we will certainly implement and discuss many of their suggestions in future work.

Sincerely,

The authors

---

### Decision · Program_Chairs · 2020-11-02

Accept (Oral)